# Succinate Injection Rescues Vasculature and Improves Functional Recovery Following Acute Peripheral Ischemia in Rodents: A Multimodal Imaging Study

**DOI:** 10.3390/cells10040795

**Published:** 2021-04-02

**Authors:** Anaïs Moyon, Philippe Garrigue, Laure Balasse, Samantha Fernandez, Pauline Brige, Ahlem Bouhlel, Guillaume Hache, Françoise Dignat-George, David Taïeb, Benjamin Guillet

**Affiliations:** 1Aix Marseille University, INSERM 1263, INRAE 1260, C2VN, 13005 Marseille, France; anais.moyon@ap-hm.fr (A.M.); philippe.GARRIGUE@univ-amu.fr (P.G.); laure.BALASSE@univ-amu.fr (L.B.); Ahlem.bouhlel@univ-amu.fr (A.B.); guillaume.hache@univ-amu.fr (G.H.); Francoise.dignat-george@univ-amu.fr (F.D.-G.); 2Aix-Marseille University, CNRS 2012, CERIMED, 13005 Marseille, France; samantha.fernandez@univ-amu.fr (S.F.); Pauline.Brige@univ-amu.fr (P.B.); david.taieb@ap-hm.fr (D.T.); 3APHM, Service de Radiopharmacie, 13005 Marseille, France; 4Aix-Marseille University, UR4264, LIIE, 13005 Marseille, France; 5APHM, Service d’Hématologie, Hôpital Conception, 13005 Marseille, France; 6Aix-Marseille University, INSERM 1068, CNRS 7258, CRCM, 13008 Marseille, France; 7APHM, Service de Médecine Nucléaire, 13005 Marseille, France

**Keywords:** succinate, RGD, gallium-68, PET, angiogenesis, vascularization, GPR91

## Abstract

Succinate influences angiogenesis and neovascularization via a hormonelike effect on G-protein-coupled receptor 91 (GPR91). This effect has been demonstrated in the pathophysiology of diabetic retinopathy and rheumatoid arthritis. To evaluate whether succinate can play a role in acute peripheral ischemia, a preclinical study was conducted with ischemic mice treated with succinate or PBS and evaluated by imaging. Acute ischemia was followed by an increased in GPR91 expression in the ischemic muscle. As assessed with LASER-Doppler, succinate treatment resulted in an earlier and more intense reperfusion of the ischemic hindlimb compared to the control group (* *p* = 0.0189). A microPET study using a radiolabeled integrin ligand ([^68^Ga]Ga-RGD_2_) showed an earlier angiogenic activation in the succinate arm compared to control mice (* *p* = 0.020) with a prolonged effect. Additionally, clinical recovery following ischemia was better in the succinate group. In conclusion, succinate injection promotes earlier angiogenesis after ischemia, resulting in a more effective revascularization and subsequently a better functional recovery.

## 1. Introduction

Succinate is an intermediate of the tricarboxylic acid cycle (e.g., Krebs cycle), and is oxidized to fumarate by the succinate dehydrogenase complex (SDH), which also belongs to the electron transport chain [1]. Beyond its role in carbohydrate metabolism, succinate was found to play an oncometabolite role in SDH-mutated pheochromocytomas and paragangliomas [2]. Additionally, succinate has hormone-like actions in various conditions, including malignancies [3]. These effects are related to paracrine effects via an extracellular G-protein-coupled receptor named succinate receptor 1 (SUCNR1/GPR91) [4]. Cancer promoting effects of succinate-GPR91 signaling have recently been recognized, and include induction of epithelial to mesenchymal transition, migration, and metastatic spread of lung cancer cells as well as immunosuppressive effects [5]. Involvement of GPR91 in tumor angiogenesis has also been proposed [6]. GPR91 is expressed in several tissues such as the spleen, the kidneys, the liver, the placenta, the myocardium, the retina, and the aorta [7,8,9].

Succinate acts as a signaling molecule in the cardiovascular system. Intravenous administration of succinate increases plasma renin activity and mean arterial blood pressure in rats [4]. In cardiomyocytes, succinate increases global Ca^2+^ transient, an effect that is mediated by GPR91 receptors and PKA activation [8]. Regarding its vascular effects, succinate acts as a mediator of vessel growth in normal retinal development and is involved in proliferative ischemic retinopathy via increased production of angiogenic factors, such as vascular endothelial growth factor (VEGF) [7]. In rats, the expression of GPR91 was previously described in vascular endothelial cells, in both terminal afferent arteriole and glomerulus [10]. More recently, Diehl et al. demonstrated that GPR91 was expressed in the mouse aorta [9].

Focusing on angiogenesis regulation, we recently reported that succinate enhanced [^18^F]fluorodeoxyglucose ([^18^F]FDG) uptake, a radiolabeled glucose analogue, by endothelial cells [11]. During ischemia, when oxidative phosphorylation is blocked, succinate is increased via the reverse action of SDH. Recent studies have shown that activation of the GPR91-ERK1/2-C/EBP-β signaling pathway plays an important role in regulation of angiogenesis through activation of VEGF expression [6] and was associated with neovascularization in diabetic retinopathy [12] and rheumatoid arthritis [13].

Although in vitro and in vivo studies have shown that interplay between succinate, GPR91 and proangiogenesis, the potential effect of succinate injection in postischemic conditions has not been studied. To this aim, we have evaluated the effects of succinate injection in a mouse model of hindlimb ischemia by preclinical imaging (angiogenesis PET/CT imaging and LASER-Doppler imaging). We have also studied the effect of ischemia on GPR91 expression.

## 2. Materials and Methods

### 2.1. Animals

Twelve-week-old Swiss female mice (*n* = 28; Janvier Labs, Le Genest-Saint-Isle, France) were housed in enriched cages placed in a temperature- and hygrometry-controlled room with daily monitoring and fed with water and commercial diet ad libitum.

### 2.2. Mouse Model of Hindlimb Ischemia

Unilateral hindlimb ischemia was induced as previously described [14]. Briefly, mice underwent ligation and partial resection of the right femoral artery under isoflurane anesthesia (induction at 4% in air, then 1.5% in air; Isovet from Piramal, Voorschoten, Netherlands) on a 37 °C-heated bed. Experimental surgery was performed by a trained operator.

### 2.3. Succinate Administration

Ischemic mice were split in two different treatment groups: 15 mice were daily injected in the right quadriceps femoris muscle with 10 μL of a 1 nmol μL^−1^ solution of pH 7.4 succinate (Merck, St. Quentin Fallavier, France) diluted in phosphate buffer saline (PBS, Merck). Another group of 13 ischemic mice was injected in the right quadriceps femoris muscle with 10 µL of PBS. Daily succinate and PBS injections points were performed for 21 days. Daily injections were performed in different injection spots within the quadriceps muscle of the ischemic limb.

### 2.4. Immunohistochemistry

To evaluate tissue GPR91 succinate receptor expression, two subgroups of 5 succinate-treated mice and 5 PBS-treated mice were euthanized on day 4 after ischemia. Ischemic and contralateral gastrocnemius muscles were harvested and fixed in 10% neutral-buffered formalin and embedded in paraffin (Tissue-Tek VIP5 Jr, Sakura, Villeneuve d’Ascq, France). Five micrometer-thick sections were stained with a GPR91 NPB1-00861 (Novus Biologicals, Centennial, CO, USA) using a fully automated BOND-III stainer (Leica) and a Novocastra Bond Polymer Refine Detection Kit (Leica) containing a postprimary peroxidase-blocking polymer reagent and a diaminobenzidine chromogen, and counterstained with hematoxylin. Slides were analyzed under an optical microscope (Eclipse, Nikon Healthcare, Champigny sur Marne, France) and results were expressed as GPR91 intensity/surface (µm^2^).

### 2.5. Immunofluorescence

To evaluate colocalization of GPR91 expression on endothelial cells, we performed immunofluorescence staining in 5 succinate-treated mice euthanized on day 4 after ischemia. Ischemic and contralateral gastrocnemius muscles were submitted to the same treatment as for immunohistochemistry. After antigenic unmasking and nonspecific protein binding blocking, thick sections were manually costained with GPR91 NPB1-00861 (Novus Biologicals) and with isolectin GS-IB4 Fluor 488 121,411 (Life technologies, Carlsbad, CA, USA). Then sections were incubated with goat anti-rabbit secondary antibody, Fluor 647 (Life technologies), after appropriate incubation and final PBS wash, each section was mounted with DAPI-Fluoromount solution (Thermo-Fisher Scientific, Waltham, MA, USA). Isolectin B4 and GPR91 expressions were evaluated using a microscope (Eclipse, Nikon Healthcare, Champigny sur Marne, France).

### 2.6. Western Blot

Western blot analysis was performed as previously described [14]. Briefly, mice were euthanized on day 4 postischemia. Gastrocnemius ischemic and contralateral muscles of PBS- and succinate-treated groups were harvested and directly snap-frozen in liquid nitrogen. Each muscle was mechanically milled in cold lysis buffer (Tris pH 8, 10 mmol L^−1^, EDTA 1 mmol L^−1^ pH 8, NaCl 150 mmol L^−1^, and NP40 10% 1 mL). Proteins were quantified by protein assay (BCA protein assay kit; Pierce). Thirty micrograms of protein were mixed with NuPAGE lithium dodecylsulfate sample buffer (Invitrogen, Carlsbad, CA, USA) and NuPAGE sample reducing agent (Invitrogen). Samples were then subjected to NuPAGE using 4–12% Novex Bis-Tris gels (Invitrogen), and separated proteins were transferred onto nitrocellulose membranes (Invitrogen, Carlsbad, CA, USA). Membranes were probed with specific primary antibodies: anti-GPR91 antibody NPB1-00861 (Novus Biologicals) incubated overnight with a control; rabbit anti-actine D6A8 antibody (Cell Signaling Technology, Danvers, MA, USA). Membranes were then washed three times with Tris-Buffered Saline/Tween-20 solution (TBST, Tris-HCL pH 7.6 0.05 mmol L^−1^, NaCl 0.3 mmol L^−1^, Tween-20 0.1%) and incubated with respective peroxidase-coupled goat anti-rabbit secondary antibody: 31,460 (Thermo-Fisher Scientific). Blots were revealed with the ECL substrate Pierce (Thermo-Fisher Scientific). Results were expressed as a ratio of GPR91 intensity in ischemic muscle to GPR91 intensity in control muscle, both previously corrected by actin intensity.

### 2.7. [^68^Ga]Ga-RGD_2_ Radiosynthesis

Gallium-68 chloride ([^68^Ga]GaCl_3_, 200  ±  41 MBq/500 µL) was obtained from the elution of a commercial TiO_2_-based [^68^Ge]Ge/[^68^Ga]Ga generator (Galliapharm, Eckert & Ziegler, Berlin, Germany) using 0.1 N HCl, buffered with a fresh 4 mol L^−1^ ammonium acetate solution (pH 7.4) and mixed with 10 μg of arginine-glycine-aspartate (RGD) dimer acetate precursor (RGD_2_, ABX Chemicals, Radeberg, Germany), reaching a final pH of 6.0. The reaction mixture was stirred at room temperature for 10 min with gentle agitation. Radiochemical purity was determined by radio-thin layer chromatography (TLC) using a miniGITA radio-TLC scanner detector (Elysia-Raytest, Straubenhardt, Germany), iTLC-SG paper as solid phase (Agilent, Santa Clara, CA, USA), a 1:1 (*v*/*v*) mixture of 1 mol L^−1^ aqueous ammonium acetate solution, and absolute methanol as mobile phase 1 (*R*_f_ [free ^68^Ga]/[^68^Ga]Ga-RGD_2_: 0/1) and a 0.1 mol L^−1^ sodium citrate pH 5 solution as mobile phase 2 (*R*_f_ [^68^Ga]-RGD_2_/[free ^68^Ga]: 0/1). Radiosyntheses with radiochemical purities ≥95.0% were used.

### 2.8. [^68^Ga]Ga-RGD_2_ MicroPET/CT Imaging

Mice were injected in the caudal vein with 5.8 ± 1.5 MBq/50 µL [^68^Ga]Ga-RGD_2_ on days 4, 7, 10 and 14 after ischemia. MicroPET/CT images were acquired 60 min after injection during 20 min. MicroPET/CT imaging sessions were performed on a Nanoscan PET/CT camera (Mediso, Budapest, Hungary) under isoflurane anesthesia (induction at 4% in air, then 1.5% in air; Isovet from Piramal, Voorschoten, Netherlands). Quantitative region-of-interest (ROI) analysis of the PET signal was performed on attenuation- and decay-corrected PET images using InterviewFusion software v3.01 (Mediso) and tissue uptake values were expressed as mean± SD of percentage of the injected dose per gram of tissue (%ID/g).

### 2.9. Hindlimb Perfusion

LASER-Doppler perfusion imaging (PIM2, Perimed, Craponne, France) was used to assess hindlimb perfusion after surgery on days 1, 3, 8, 15 and 24 postischemia under isoflurane anesthesia (induction at 4% in air, then 1.5% in air) on a 37 °C-heated bed. Results were expressed as a ratio of ischemic to nonischemic hindlimb blood flow (*i*/*c* ratio).

### 2.10. Motility Impairment Score

A motility impairment score was used to evaluate individual clinical recovery on days 1, 3, 8, 15 and 24 as follows: 1, unrestricted active movement; 2, restricted active foot; 3, use of the other leg only; 4, leg necrosis; 5, self-amputation.

### 2.11. Statistical Analysis

Biodistribution data were analyzed using Prism software v8.4.2 (Graphpad, San Diego, CA, USA). Data were expressed as mean values ± SD unless indicated otherwise. Differences between ischemic and contralateral GPR91 expression in muscles were analyzed using a parametric paired Student *t*-test. Gaussian distributions were assumed by a Shapiro-Wilk normality test. Differences were considered statistically significant when *p* < 0.05. MicroPET signal quantification, LASER-Doppler perfusion and motility impairment scoring were analyzed using one-way ANOVA with a post-hoc Sidak test.

## 3. Results

### 3.1. Ischemia Is Associated with an Increase of GPR91

Representative immunohistochemistry images on day 4 in PBS- and succinate-treated mice are shown in Figure 1A. Quantitative analysis highlighted a significant overexpression of GPR91 in ischemic muscle compared to contralateral muscle in PBS-treated mice (respectively 11.9 ± 2.44 intensity/µm^2^ and 8.58 ± 3.61 intensity/µm^2^; * *p* = 0.028, *n* = 5) and in succinate-treated mice (respectively 51.6 ± 19.2intensity/µm^2^ and 20.7 ± 6.6 intensity/µm^2^; * *p* = 0.012, *n* = 5). Most interestingly, a significant overexpression of GPR91 was found in ischemic muscles of succinate-treated mice compared to these of PBS-treated mice (** *p* = 0.0018, *n* = 5; Figure 1B).

Representative Western blot images on day 4 in PBS- and succinate-treated mice are shown in Figure 1C. Quantitative analysis showed a significant overexpression of GPR91 in ischemic succinate-treated muscle compared to ischemic PBS-treated muscle (respectively 1.10 ± 0.15 and 3.40 ± 0.87 expressed as ischemic-to-contralateral ratio; ** *p* = 0.0043, *n* = 5 and *n* = 6; Figure 1D).

### 3.2. GPR91 Colocalizes with Endothelial Cells in Ischemic Muscles Treated by Succinate

Representative images of immunofluorescent staining of GPR91 and isolectin B4 on day 4 in succinate-treated mice are shown in Figure 1E. Analysis of images showed a colocalization of GPR91 expression and endothelial cells in ischemic muscles of succinate treated group.

### 3.3. Succinate Injection Induces Earlier [^68^Ga]Ga-RGD_2_ Uptake Following Hindlimb Ischemia

Representative images from PET imaging of [^68^Ga]Ga-RGD_2_ PET in both groups for day 4, 7 and 10 are shown in Figure 2A. At day 4 postischemia, a significant increase of [^68^Ga]Ga-RGD_2_ uptake was observed in ischemic muscles treated by succinate compared to ischemic muscles treated by PBS, and compared to contralateral muscles (PBS: 0.10 ± 0.07%ID/g, *n* = 8; succinate: 0.29 ± 0.19%ID/g, *n* = 10; * *p* = 0.020; contralateral: 0.09 ± 0.03%ID/g, *n* = 10; ^###^
*p* < 0.0001). By contrast, on day 10, the increased [^68^Ga]Ga-RGD_2_ uptake was observed in ischemic muscles treated by PBS compared to ischemic muscle treated by succinate, and compared to contralateral muscles (PBS: 0.42 ± 0.23%ID/g, *n* = 8; succinate: 0.23 ± 0.14%ID/g, *n* = 10; ^&^
*p* = 0.040; contralateral: 0.20 ± 0.07%ID/g, *n* = 8; ^$$^
*p* = 0.002; Figure 2B).

### 3.4. Succinate Injection Increases Blood Perfusion Recovery Following Hindlimb Ischemia

Blood perfusion recovery was significantly earlier and more intense in succinate treatment group compared to PBS group (* *p* = 0.0189) especially on day 8 (PBS: 49.3 ± 20.8%, *n* = 8; succinate: 72.3 ± 18.5%, *n* = 10; * *p* = 0.011) and on day 15 (PBS: 50.6 ± 23.4%, *n* = 8; succinate: 75.8 ± 8.9%, *n* = 10; *** *p* = 0.004), as shown in Figure 3A,B.

### 3.5. Succinate Injection Increases Clinical Recovery Score Following Hindlimb Ischemia

Functional recovery was significantly higher in the succinate group than in the PBS group (PBS: 2.87 ± 1.12, *n* = 8; succinate: 1.50 ± 1.07, *n* = 10; ** *p* = 0.010) on day 24 postischemia (Figure 3C).

## 4. Discussion

The present study illustrates the versatile actions of succinate. We showed for the first time that succinate injection following hindlimb ischemia in mice lead to an earlier blood perfusion recovery and most importantly a better functional recovery score. The vascular effect was shown on both early angiogenesis activation assessed by [^68^Ga]Ga-RGD_2_ PET/CT and functional tissue perfusion by LASER-Doppler imaging.

[^68^Ga]Ga-RGD_2_ enables quantification of endothelium activation engaged in the process of neovessel formation. The RGD tripeptide sequence has high affinity and specificity towards the α_v_β_3_ integrin, which is overexpressed during angiogenesis and is therefore considered as a reliable angiogenesis biomarker. Indeed, α_v_β_3_ integrin expression is highly restricted on other cell types by healthy tissues [15] whereas highly expressed on activated endothelial cells of tumor neovasculature and/or by some tumor cells themselves [16]. In the postischemic context, α_v_β_3_ integrin is mainly highly expressed on activated endothelial cells, compared to its low expression on activated leucocytes. This minor expression has been described as insignificantly contributing to the uptake of RGD-harboring radiotracers, even in inflammatory preclinical models [17].

We showed on [^68^Ga]Ga-RGD_2_ PET longitudinal follow-up study that angiogenic activation peaked on day 4 following acute ischemia in succinate-treated mice and no sooner than on day 8 in PBS-treated mice, resulting in an earlier limb revascularization in succinate-treated mice, which is known to be significantly associated with restoration of muscle contractile function [18]. This is in agreement with our study that showed an improvement of functional recovery in the succinate group compared to the control group. Changes in LASER–Doppler flow patterns are delayed compared to molecular imaging with differences observed on day 8 and 15 since LASER-Doppler assesses blood flow in functional vessel. Taken together, these data suggest that succinate promotes angiogenesis, possibly resulting in better outcomes following hindlimb ischemia.

According to the literature, GPR91 seems to be implied in the activation of coping mechanisms upon adverse conditions, including stimulation of proliferation of different cell types, migration, and angiogenesis [1,7,12,19,20,21,22]. This has also been illustrated in ischemic conditions [23,24,25]. Our therapeutic scheme has probably amplified the pathophysiological response to ischemia and/or increased the bioavailability of succinate since it has been directly injected into the muscles for 21 days. We showed that GPR91 expression was increased on day 4 in muscle following ischemia. Interestingly, GPR91 expression was shown to be induced by hypoxia [26]. We assumed that the effect of succinate was related to the activation of GPR91. This hypothesis could be further studied by the disruption of GPR91 activation by targeted treatments with GPR91 antagonists.

At present, the beneficial role of succinate metabolism pathway in the ischemic cascade is still debated, notably regarding reports advocating malonate derivatives use to compete succinate degradation by SDH, supposed to inhibit the raise of mitochondrial oxygen species [27]. As such strategies have shown beneficial effects [28], these effects cannot be only explained by the decrease in oxidative stress as hypothesized by the authors, since malonate derivatives lead also to an accumulation of extracellular succinate. Moreover, from a mechanical point of view, succinate/GPR91 pathway activation may prevent the degradation of hypoxia-inducible factor 1-alpha, of which the pathway is known to promote angiogenesis and cell metabolism [29].

## 5. Conclusions

Our data provide new impetus for succinate or GPR91 modulation during acute ischemia.

Succinate injection may constitute a promising new treatment approach for acute limb ischemia. Further basic research studies are needed to identify potential synergistic approaches and describe the underlying molecular mechanisms. Effect of succinate would also merit to be evaluated in the setting of with chronic peripheral arterial disease, which concerns 200 million patients worldwide with growing incidence [30].

## Figures and Tables

**Figure 1 cells-10-00795-f001:**
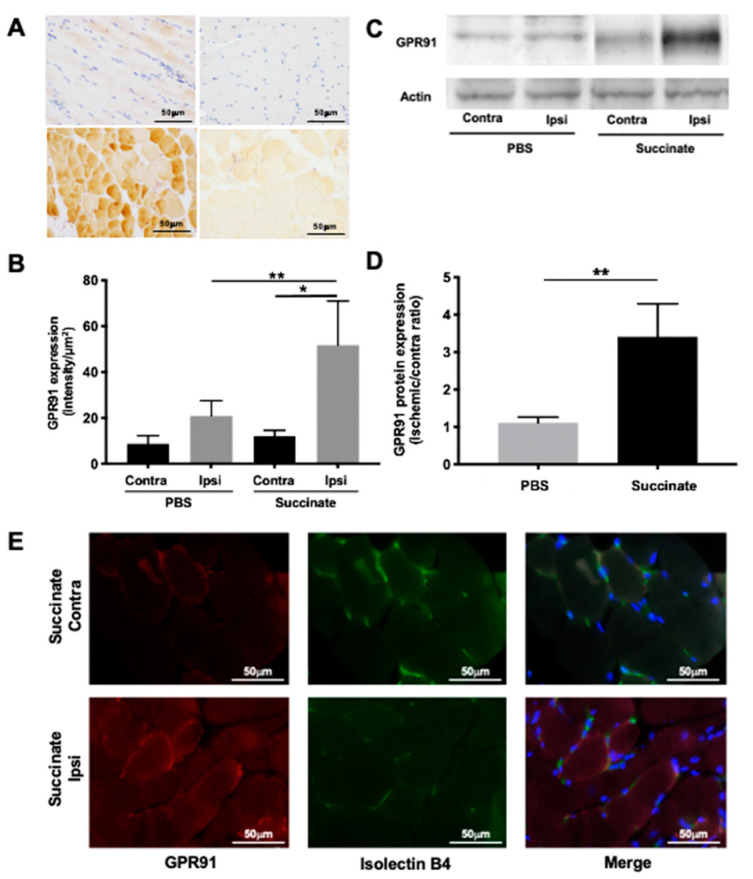
Immunohistochemistry staining of G-protein-coupled receptor 91 (GPR91) in mice gastrocnemius muscle. (**A**) Representative GPR91 immunostaining on day 4 in contralateral hindlimb phosphate buffer saline (PBS) group (a); ischemic hindlimb PBS group (b); contralateral hindlimb succinate group (c); ischemic hindlimb succinate group (d). (**B**) Quantitative analysis showed a significant overexpression of GPR91 in ischemic muscle compared to contralateral muscle in PBS-treated mice (* *p* = 0.028, *n* = 5), in succinate-treated mice (* *p* = 0.012, *n* = 5), and a significant overexpression of GPR91 in ischemic muscle of succinate-treated mice compared to that of PBS-treated mice (** *p* = 0.0018, *n* = 5). (**C**) Western blot analysis of GPR91 on gastrocnemius lysis muscle. Representative GPR91 and actin on day 4 in contralateral hindlimb PBS group, ischemic hindlimb PBS group, contralateral hindlimb succinate group and ischemic hindlimb succinate group. (**D**) Quantitative analysis showed a significant overexpression of GPR91 in succinate ischemic muscle compared to PBS ischemic muscle (** *p* = 0.0043, *n* = 5 and *n* = 6). Immunofluorescent staining of GPR91 and isolectin B4 in mice gastrocnemius muscle. (**E**) Representative GPR91 and isolectin B4 immunostaining showed colocalization of both realized on day 4 in contralateral hindlimb and ischemic hindlimb succinate-treated group.

**Figure 2 cells-10-00795-f002:**
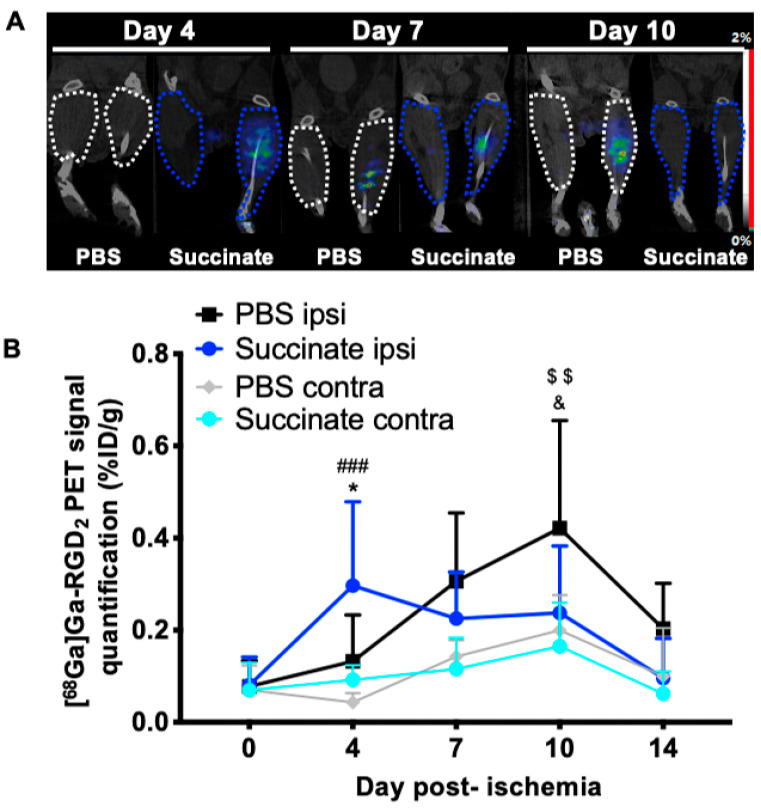
[^68^Ga]Ga-RGD_2_ microPET-CT imaging. (**A**) Representative microPET-CT imaging on days 4, 7 and 10 in PBS and succinate-treated mice for contralateral and ischemic hindlimb. (**B**) Image-derived time-activity curves from day 0 to day 14 of [^68^Ga]Ga-RGD_2_ microPET-CT imaging as a percentage of the injected dose per gram of tissue (%ID/g) (mean± SD; *n* = 8 for PBS group and *n* = 10 for succinate group; * *p* = 0.02 for day 4 and ^&^
*p* = 0.04 for day 10, between PBS and succinate ischemic hindlimb; ^###^
*p* < 0.0001 for day 4, between ischemic and contralateral succinate-treated group; ^$$^
*p* = 0.002 for day 10, between ischemic and contralateral PBS-treated group).

**Figure 3 cells-10-00795-f003:**
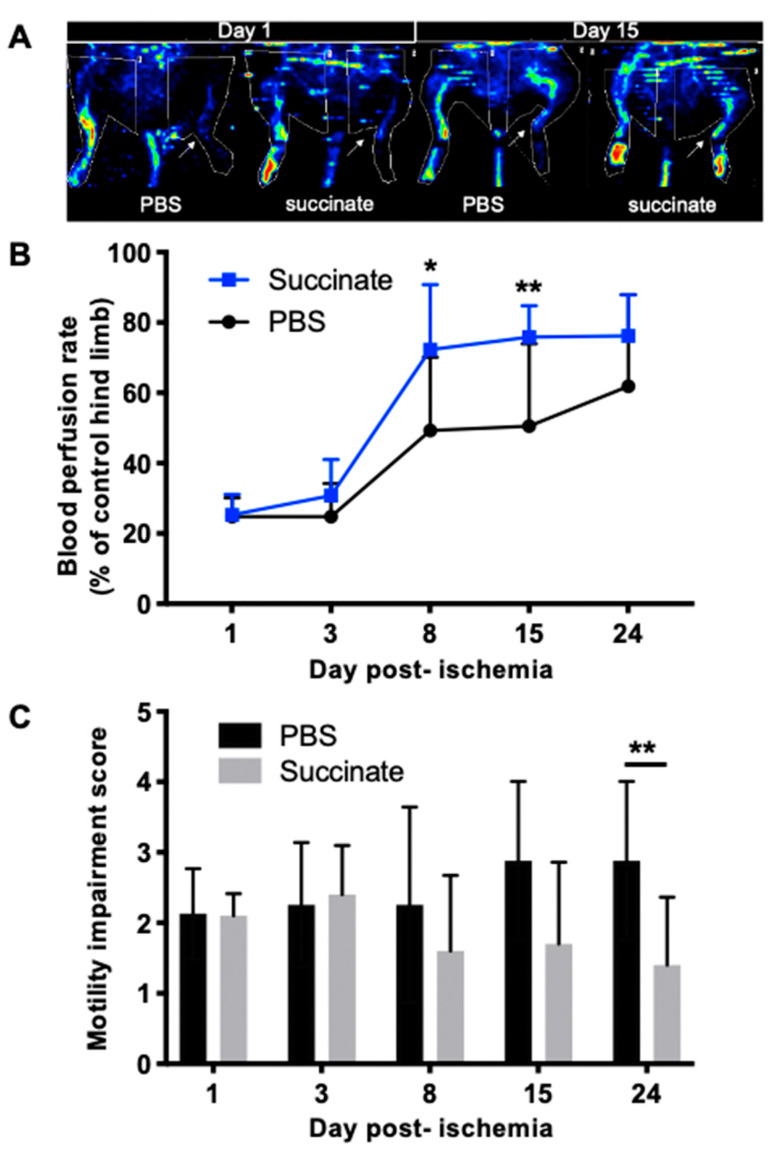
(**A**) Representative LASER-Doppler perfusion images on days 1 and 15 postischemia in PBS- and succinate-treated mice; (**B**) quantitative analysis of LASER-Doppler perfusion imaging expressed as ischemic-to-contralateral muscle signal ratio (%, mean ± SD) from day 1 to day 24 postsurgery showed a significant perfusion decrease in succinate group perfusion on days 8 and 14 compared to the PBS group (* *p* = 0.01 and ** *p* = 0.004 respectively;  *n* = 8 for PBS group and *n* = 10 for succinate group); (**C**) motility impairment score. 1: unrestricted active movement; 2: restricted active foot; 3: use of the other leg only; 4: leg necrosis; 5: self-amputation (** *p* = 0.01; *n* = 8 for PBS group and *n* = 10 for succinate group).

## Data Availability

The data presented in this study are available on request from the corresponding author.

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
