# Peer review of "Succinate Injection Rescues Vasculature and Improves Functional Recovery Following Acute Peripheral Ischemia in Rodents: A Multimodal Imaging Study"

_cells, 2021, doi:10.3390/cells10040795_

Round 1
Reviewer 1 Report
This paper described using Ga-RGD in PET imaging to examine angiogenesis in mice after hindlimb ischemia, in comparison with conventional laser Doppler method. Authors also showed interesting and novel result of succinate treatment improves perfusion recovery. These findings are novel and important. The manuscript is well written and pleasant to read. Below are some of my comments and suggestions.
- Authors chose the Swiss mice, which is not a common strain to study HI. Most of the published literatures are based on Balb/c or C57BL/6. Please provide some justification of using this strain.
- Please specify gender.
- Please provide more details for the succinate intramuscular injection, specifically whether authors injected at different spots or the same spot every day for the total of 21 days.
- The GPR91 expression clearly upregulated in response to succinate treatment. Authors are suggested to elaborate more on whether this is associated with specific type of cells in the muscle. Therefore, if possible, authors can provide co-stain of GPR91 with cell type markers (muscle fiber, satellite cells, endothelial, smooth muscle cells, for example).
- If possible, authors can consider to show one more later time point of GPR91 expression after ischemia.
- The target of RGD, integrin αvβ3, is mostly expressed on angiogenic endothelial cells, and also on activated/inflamed endothelial cells. Earlier increase of Ga-RGD signal could be due to not only angiogenesis. However, Doppler flow only measures the functional recovery. If possible, authors can discuss more on the limitations as well.
Author Response
Dear Reviewer,
We thank you very much for your valuable comments. Please find below our point-by-point response to your requests.
All the modifications resulting from this reviewing have been highlighted in yellow in the manuscript.
We did everything we could to perform the requested experiments within the 10-day reviewing deadline.
1. Authors chose the Swiss mice, which is not a common strain to study HI. Most of the published literatures are based on Balb/c or C57BL/6. Please provide some justification of using this strain.
Due to the COVID condition and its impact on animal availability, we decided to switch from Balb/c to Swiss strain for HLI model in our laboratory. Even if there are strain specificities in ischemic-induced injury (Helisch et al., 2006 ATVB : https://doi.org/10.1161/01.ATV.0000202677.55012.a0 ), Swiss strain is entirely eligible for the realization of the HLI model as reported in the literature (Lejay et al., 2015 EJVES : https://doi.org/10.1016/j.ejvs.2014.12.010 ).
2. Please specify gender
As requested by the reviewer, the gender was added to the manuscript (female mice). Indeed, the gender might influence long-term revascularization prognosis after critical limb ischemia in patients (Lejay et al., 2015 EJVES : https://doi.org/10.1016/j.ejvs.2015.07.014 ).
Either sex can be used for the scientific purpose of HLI study (Padgett et al., 2016 J Vis Exp : https://doi.org/10.3791/54166 ). Facing both increase in smoking and ischemia-related diseases prevalence in women, we decided in our lab to study specifically ischemic diseases in female (peripheral artery disease and stroke models).
3. Please provide more details for the succinate intramuscular injection, specifically whether authors injected at different spots or the same spot every day for the total of 21 days.
As requested by the reviewer, we have specified in the manuscript that daily injections were performed in different injection spots within the quadriceps muscle of the ischemic limb.
4. The GPR91 expression clearly upregulated in response to succinate treatment. Authors are suggested to elaborate more on whether this is associated with specific type of cells in the muscle. Therefore, if possible, authors can provide co-stain of GPR91 with cell type markers (muscle fiber, satellite cells, endothelial, smooth muscle cells, for example).
As requested by the reviewer we realized colocalization of GPR91 receptor and endothelium by immunofluorescence staining study using GPR91 antibody and isolectin B4.
We have specified in the manuscript materials & methods and results of this experiment. Representative images of immunofluorescent staining of GPR91 and isolectin B4 on day 4 in succinate-treated mice are shown in Figure 1C. Analysis of images showed colocalization of GPR91 receptors expression and endothelial cells in ischemic muscle of succinate treated group.
5. If possible, authors can consider to show one more later time point of GPR91 expression after ischemia.
Later time points of GPR91 expression assessments are clearly of interest but need additional animals and experiments, which we were unfortunately unable to perform within the allotted 10-day deadline for reviewed manuscript submission.
6. The target of RGD, integrin αvβ3, is mostly expressed on angiogenic endothelial cells, and also on activated/inflamed endothelial cells. Earlier increase of Ga-RGD signal could be due to not only angiogenesis. However, Doppler flow only measures functional recovery. If possible, authors can discuss more on the limitations as well.
According to the reviewer’s comment, this has been added to the Discussion section as follows :
“Indeed, αvβ3 integrin expression is highly restricted on other cell types by healthy tissues (Eliceiri BP et al., 1999 J Clin invest) whereas highly expressed on activated endothelial cells of tumor neovasculature and/or by some tumor cells themselves (Zhou et al., 2011 Theranostics). In the post-ischemic context, αvβ3 integrin is mainly highly expressed on activated endothelial cells, compared to its low expression on activated leucocytes. This minor expression has been described as unsignificantly contributing to the uptake of RGD-harboring radiotracers, even in inflammatory preclinical models (Pichler et al., 2005 JNM).”
Reviewer 2 Report
The manuscript by Moyon et al. examined effect of succinate on acute peripheral ischemia in vivo. Treatment with succinate showed an earlier angiogenic activation and increase in clinical recovery score following hindlimb ischemia. Although the detailed mechanism is still uncovered, these findings of the study are significant. I have some comments to the authors.
Comment 1: In Figure 1, immunohistochemistry staining showed GPR91 expression in gastrocnemius muscle was significantly increased in ischemic hindlimb groups compared to control groups (both PBS and Succinate). For quantification measurement, can you confirm the results by Western blotting or/and qPCR?
Also, do the authors examine whether GPR91 is expressed in vascular endothelium and succinate injection increases GPR91 expression in vascular endothelium?
Comment 2: In Figure 2, please put a scale bar of intensity. In addition, did the authors observe increase in angiogenesis in gastrocnemius muscle of succinate injection group compared to control group (Figure 1). If yes, would you please show the result?
Comment 3: In Figure 3, succinate injection increased blood perfusion and clinical recovery score in ischemia mice. Are there any effects of succinate injection on control mice?
Author Response
Dear Reviewer,
We thank you very much for your valuable comments. Please find below our point-by-point response to your requests.
All the modifications resulting from this reviewing have been highlighted in yellow in the manuscript.
We did everything we could to perform the requested experiments within the 10-day reviewing deadline.
Comment 1: In Figure 1, immunohistochemistry staining showed GPR91 expression in gastrocnemius muscle was significantly increased in ischemic hindlimb groups compared to control groups (both PBS and Succinate). For quantification measurement, can you confirm the results by Western blotting or/and qPCR?
As requested by the reviewer, we have confirmed immunohistochemistry results by Western Blot. Materials&methods and Results sections have been updated in the manuscript accordingly. Representative Western blot images on day 4 in PBS- and succinate-treated mice are shown in Figure 1D. Quantitative analysis showed significant overexpression of GPR91 in ischemic succinate-treated muscle compared to ischemic PBS-treated muscle (respectively 1.10±0.15 and 3.40±0.87 expressed on ischemic to contralateral ratio; **P=0.0043, n=5 and n=6) Figure 1E.
Also, do the authors examine whether GPR91 is expressed in vascular endothelium and succinate injection increases GPR91 expression in vascular endothelium?
As requested by the reviewer, we assessed colocalization of GPR91 receptor and endothelium by immunofluorescence staining study using GPR91 antibody and isolectin B4.
Materials&methods and Results sections have been updated accordingly. Representative images of immunofluorescent staining of GPR91 and isolectin B4 on day 4 in succinate-treated mice are shown in Figure 1C. Analysis of images showed colocalization of GPR91 receptors expression and endothelial cells in ischemic muscle of the succinate-treated group.
Comment 2: In Figure 2, please put a scale bar of intensity.
In addition, did the authors observe an increase in angiogenesis in the gastrocnemius muscle of the succinate injection group compared to control group (Figure 2). If yes, would you please show the result?
Scale bar of intensity and %ID/g of the control group have been added to Figure 2 as requested
and corresponding statistical differences have been implemented within Figure 2 and Results section as follows:
“Representative images from PET imaging of [68Ga]Ga-RGD2 PET in both groups for day 4, 7 and 10 are shown in Figure 2A. At day 4 post-ischemia, a significant increase of [68Ga]Ga-RGD2 uptake was observed in ischemic muscles treated by succinate compared to ischemic muscles treated by PBS and to contralateral muscles (PBS: 0.10±0.07%ID/g, n=8; succinate: 0.29±0.19%ID/g, n=10; *P=0.020; contralateral: 0.09±0.03%ID/g, n=10; ###P<0.0001). By contrast, on day 10, the [68Ga]Ga-RGD2 increased uptake was observed in ischemic muscles treated by PBS compared to ischemic muscle treated by succinate and to contralateral muscles (PBS: 0.42±0.23%ID/g, n=8; succinate: 0.23±0.14%ID/g, n=10; &P=0.040; contralateral: 0.20±0.07%ID/g, n=8; $$P=0.002, Figure 2B)”.
Comment 3: In Figure 3, succinate injection increased blood perfusion and clinical recovery score in ischemia mice. Are there any effects of succinate injection on control mice?
Succinate injection with the same administration scheme in control mice would effectively have been of interest but would have needed additional animals and experiments, which we were unfortunately unable to perform within the allotted 10-day deadline for reviewed manuscript submission. Of note, our previously published works (Garrigue et al. JNM 2017) evaluated the acute effect of succinate vs PBS injections every 6h for 24h in gastrocnemius muscles of healthy mice and found no significant difference on LASER-Doppler perfusion 28h after the last injection.
Round 2
Reviewer 2 Report
The authors of the manuscript address the reviewer’s comments. I do not have any more comments for this.